# Protective effects of ectoine on articular chondrocytes and cartilage in rats for treating osteoarthritis

Peng Li[1,2☯]*, Yong Huang[2☯], Lishuai Miao[2], Zhiqi Zhu[2], Zhanjun Shi[1]*

1 Department of Orthopedic Surgery, Nanfang Hospital, Southern Medical University, Guangzhou, Guangdong, China, 2 Orthopedic Surgery Department of The Second Affiliated Hospital, School of Medicine, The Chinese University of Hong Kong, Shenzhen & Longgang District People's Hospital of Shenzhen, Shenzhen, Guangdong, China

☯ These authors contributed equally to this work.
* lipeng8436@126.com (PL); 59114835@qq.com (ZS)

**Data Availability Statement:** All relevant data are within the manuscript and its Supporting information files.

**Funding:** The work was supported by Shenzhen Longgang District Medical Science and Technology

## Abstract

Osteoarthritis (OA) is a chronic degenerative disease that primarily includes articular cartilage destruction and inflammatory reactions, and effective treatments for this disease are still lacking. The present study aimed to explore the protective effects of ectoine, a compatible solute found in nature, on chondrocytes in rats and its possible application in OA treatment. In the *in vitro* studies, the morphology of the chondrocytes after trypsin digestion for 2 min and the viability of the chondrocytes at 50°C were observed after ectoine treatment. The reactive oxygen species (ROS) levels in chondrocytes pretreated with ectoine and post-stimulated with $H_2O_2$ were detected using an ROS assay. Chondrocytes were pretreated with ectoine before IL-1β stimulation. RT–qPCR was used to measure the mRNA levels of cyclooxygenase-2 (COX-2), metallomatrix proteinase-3, -9 (MMP-3, -9), and collagen type II alpha 1 (Col2A1). In addition, immunofluorescence was used to assess the expression of type II collagen. The *in vivo* effect of ectoine was evaluated in a rat OA model induced by the modified Hulth method. The findings revealed that ectoine significantly increased the trypsin tolerance of chondrocytes, maintained the viability of the chondrocytes at 50°C, and improved their resistance to oxidation. Compared with IL-1β treatment alone, ectoine pretreatment significantly reduced COX-2, MMP-3, and MMP-9 expression and maintained type II collagen synthesis in chondrocytes. *In vivo*, the cartilage of ectoine-treated rats exhibited less degeneration and lower Osteoarthritis Research Society International (OARSI) scores. The results of this study suggest that ectoine exerts protective effects on chondrocytes and cartilage and can, therefore, be used as a potential therapeutic agent in the treatment of OA.

## Introduction

Osteoarthritis (OA) is a common chronic degenerative disease of the joints. The pathological changes involved in OA are characterized by degeneration and injury of the articular cartilage

Project [grant no. LGWJ 2021-037]. The funders had no role in study design, data collection and analysis, decision to publish, or preparation of the manuscript.

**Competing interests:** The authors have declared that no competing interests exist.

[1]. Owing to the absence of blood vessels, lymph nodes, and nerve tissue and the limited ability of these vessels to regenerate, articular cartilage has a weak ability to self-repair after injury. Although debridement and lavage, microfracture, and autologous chondrocyte implantation can treat diseased cartilage, these methods have limitations. Debridement and lavage do not promote cartilage regeneration. Microfracture is a surgical technique that creates small fractures in the bone underlying cartilage injury to stimulate the growth of cartilage-like tissue. However, the resulting tissue is often fibrocartilage, which is not as strong or durable as normal articular cartilage. Autologous chondrocyte implantation is expensive and time-consuming and requires cell harvesting from a healthy part of the joint, which can lead to further damage. These conditions eventually lead to OA and seriously affect patients' quality of life owing to limb movement disorders and disabilities [2–4]. Currently, no radical treatment is available for OA. Symptomatic treatment is mainly performed through anti-inflammatory and analgesic drugs, and joint replacement is ultimately performed in the late stage. However, these treatments are restricted by issues such as short-term symptom improvement, poor long-term effects, postoperative infection, and prosthesis loosening [5]. Therefore, articular cartilage repair faces significant challenges, and new and effective therapeutic approaches are urgently needed.

Ectoine (Ec), 1,4,5,6-tetrahydro-2-methyl-4-pyrimidinecarboxylic acid, with the molecular formula $C_6H_{10}N_2$ (Fig 1A), is a compatible water molecule-binding solute (osmoprotectant) produced by several bacterial species in response to osmotic stress and unfavorable environmental conditions [6]. Galinski first discovered and isolated ectoine from *Ectothiorhodospira halochloris* in 1985. Ectoine, an amino acid derivative, is produced in the cells of several halophilic microorganisms to maintain osmotic pressure balance [7]. Ectoine does not interfere with cellular processes, can accumulate at high concentrations, and prevents cell dehydration [8]. In addition, it improves the hydration of the cell surface, increases the intramolecular distance, and improves the fluidity of the lipid head groups in the cell membrane to bind intensively with water molecules and stabilize macromolecules (Fig 1B). Ectoine protects against the biological activities of enzymes, nucleic acids, cell membranes, and other biological macromolecules in extreme environments, such as high temperature, freezing, drying, radiation, and

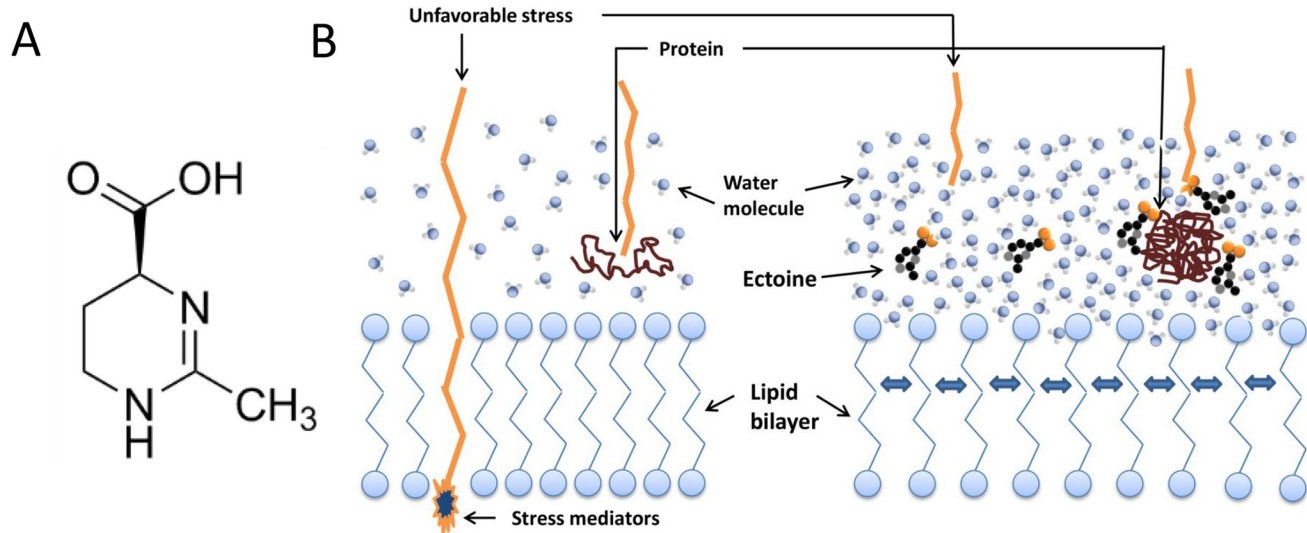

**Fig 1.** (A) Structure of ectoine. (B) Ectoine improves the fluidity of the lipid head groups in the cell membrane and stabilizes macromolecules.

free radicals. Therefore, ectoine can be widely used in the fine chemical, biomedicine, and bio-manufacturing industries [9, 10]. Currently, ectoine is used in treating diseases such as upper airway inflammation, inflammatory bowel disease, xerophthalmia, dry rhinitis, Alzheimer's disease, and Parkinson's disease. Ectoine has gained wide commercial use, particularly in the field of cosmetics [11–13].

However, no reports on the effects of ectoine on chondrocytes are available. Therefore, the effects of ectoine on digestive enzyme resistance, high-temperature resistance, and antioxidation in chondrocytes were studied. In addition, *in vitro* and *in vivo* OA models were established by interleukin (IL)-1β stimulation and the modified Hulth method to study the protective effects of ectoine on chondrocyte injury caused by OA in rats. Its potential applications in cartilage repair and OA treatment were further analyzed. This study suggested that ectoine could be beneficial in the field of cartilage repair and OA treatment in several ways. First, ectoine has anti-inflammatory properties that can help reduce cartilage damage caused by inflammation. Second, it can help protect chondrocytes from damage caused by oxidants and reactive oxygen species, which are known to play a role in OA development. Finally, ectoine promoted cartilage matrix synthesis and helped restore the structure and function of cartilage tissue.

## Materials and methods

### Animals

A total of 46 S-D rats (18 males and 18 females; aged 8 weeks (n = 40) and 4 weeks (n = 6)) were obtained from Zhuhai Bestest Biotechnology Co. (production license: SYNK(Y)2020-0230). The rats were raised in the experimental animal center of Longgang District People's Hospital of Shenzhen. The room temperature was set to 23 ±2°C with a 12 h light/dark cycle. The relative humidity was 40-60%. Food and water were provided *ad libitum*. Animal health and behavior were monitored every day. The animal experimental protocol used in this study was approved by the Ethics Committee of the Longgang District People's Hospital of Shenzhen (ethics record number: 2021005DW) and followed the "Guidelines for the Care and Use of Laboratory Animals" issued by the China Animal Research Council to minimize suffering.

### Isolation and culture of chondrocytes

Six 4-week-old S-D rats (3 males and 3 females; Zhuhai Bestest Biotechnology Co. Ltd., China) were euthanized by the inhalation of an overdose of 5% isoflurane. Briefly, cartilage was isolated from the knee joint of rats under aseptic conditions. Cartilage samples were cut into pieces, digested in 0.25% trypsin (C0201; Beyotime, China) at 37°C for 30 min, and then digested in 0.2% type II collagenase (C222; Sigma, USA) for 1 h at 37°C. Thereafter, the cells were filtered through a 200 mesh screen and centrifuged at 500 ×g for 8 min. Then, the cells were resuspended in DMEM containing 10% fetal bovine serum (16140089, Gibco, USA) and 1% penicillin/streptomycin antibiotics (100 X, C022, Beyotime, China) and cultured in a 5% $CO_2$ cell incubator at 37°C. Trypsin (0.25%) was added to the chondrocytes when the cell density reached 80%. After centrifugation at 200 × g for 5 min, the cells were passaged at a ratio of 1:3. The second-passage chondrocytes were subjected to the following tests.

### Preparation of the ectoine solution

Briefly, 50, 100, and 150 mg of ectoine (purity > 99%, SIYOMICRO, China) were weighed, added to 10 mL of culture medium, and shaken until completely dissolved. Three

concentrations of 0.5%, 1.0%, and 1.5% (*w/v*) ectoine solution were obtained by filtration through 0.22 μm sterile filters (Millipore, USA).

## Cell morphological changes

The second-passage chondrocytes were resuspended, seeded in 12-well plates ($5 \times 10^4$ cells/mL), and cultured in a 5% $CO_2$ incubator at 37˚C. When the cells had grown to a density of 50%, 1 mL of 0.5%, 1.0%, or 1.5% (*w/v*) ectoine was added to the wells. Cells not treated with ectoine were used as a blank control group. Three duplicate wells were used for each group. When the cell density reached 80%, 50 μL of 0.25% trypsin was added to digest the cells for 2 min. Finally, the cell morphology was observed under an inverted microscope (CKX53, Olympus, Japan).

## Cell viability using the MTT assay

The MTT assay was used to assess the effect of ectoine on cell viability following the manufacturer's instructions. The chondrocytes were seeded in 96-well plates at a density of 5000 cells/well. When the cells reached a density of 50%, 100 μL of 0.5%, 1.0%, or 1.5% (*w/v*) ectoine solution was added to the wells. Cells not subjected to ectoine treatment were used as the experimental control group. Six duplicate wells were used for each group. Simultaneously, zero-well (MTT, medium) and control well (chondrocytes without ectoine treatment, MTT, medium) were used. After the cells reached 80% confluence, the plants in the ectoine and experimental control groups were incubated at 37˚C and 50˚C for 1 h, respectively, while the control wells were incubated at 37˚C only. The cells were subsequently incubated with 10 μL of MTT (5 mg/ml; Beyotime, China) at 37˚C for 4 h, after which 100 μL of DMSO (D8418; Sigma, USA) was added to each well, and the mixture was shaken for 10 min. The OD of each well was measured at 490 nm using a microplate reader (SYNERH1, BioTek, USA). The viability of the chondrocytes in each group was calculated according to the following formula: viability (%) = (OD experiment–OD zero)/(OD control–OD zero) ×100%.

## ROS assay

A reactive oxygen species (ROS) assay was used to assess the effect of ectoine on antioxidation. The chondrocytes were seeded in 24-well plates ($5 \times 10^4$ cells/mL). When the cells reached 50% confluence, 0.5 mL of 0.5%, 1.0%, or 1.5% (*w/v*) ectoine solution was added to the plates, and the 0.5 mg/mL dexamethasone group was used as a positive control. Moreover, blank control and experimental control groups were established. Three duplicate wells were used for each group. When the cell density reached 80%, all the groups except for those in the blank control group were treated with 100 μM $H_2O_2$ (H1009; Sigma, USA) for 3 h. The ROS assay was performed using a ROS kit (S0033S; Beyotime, China): 2',7'-dichlorofluorescein dihydrodiacetate (DCFH-DA) was diluted 1:1000 to a final concentration of 10 μM. Thereafter, 300 μL of diluted DCFH-DA was added to each group and incubated at 37˚C for 20 min. After incubation, the cells were washed thrice to remove the DCFH-DA that did not fully enter the cells. To observe the chondrocytes, fluorescence microscopy (CKX-FL-1, OLYMPUS, Japan) was performed at excitation and emission wavelengths of 488 and 525 nm, respectively. Fluorescence intensity was measured using ImageJ software 1.5 (National Institutes of Health, USA).

## *In vitro* and *in vivo* OA models

For *in vitro* studies, chondrocytes were seeded in 6-well plates ($5 \times 10^4$ cells/mL). The cells were collected and subjected to a ROS assay. After the cells reached 80% confluence, all the

groups except for those in the blank control group were treated with IL-1β (10 ng/mL; I2393; Sigma, USA) at 37°C for 24 h. In the *in vivo* study, the OA model was generated using the modified Hulth method (for resection of the medial collateral ligament, excision of the medial meniscus, and anterior cruciate ligament), as described in a previous study [14]. The rats were anesthetized by inhalation of 1.5% isoflurane and randomized into four groups: the sham, vehicle, ectoine, and hyaluronic acid (HA) groups (n = 10/group). The HA group was used as a positive control. Starting from the third week after surgery, 50 μL of 1.5% ectoine or HA was administered once a week by intra-articular injection for four consecutive weeks. Meanwhile, the sham and vehicle group rats were treated with the same dose of physiological saline. None of the rats died during the experiment. All 40 rats were sacrificed at 8 w postsurgery by the inhalation of 5% isoflurane for more than 1 min. The absence of a heart beat and breathing were observed by touching the rat's chest for 5 min to confirm death. The 40 knee joint specimens were harvested for further historical assessment.

## RT–qPCR

In *in vitro* studies, RT–qPCR was used to detect the relative expression of the inflammatory cytokines cyclooxygenase-2 (COX-2), metallomatrix proteinase-3 (MMP-3), metalloproteinase-9 (MMP-9), and collagen type II alpha 1 (Col2A1). Total RNA was extracted following the Trizol method using Total RNA Extraction Reagent (R401, Vazyme, China). cDNA was synthesized using the PrimeScript™ RT reagent Kit (RR037A, Takara, Japan). Quantitative real-time PCR was performed following the manufacturer's instructions for Taq Pro Universal SYBR qPCR Master Mix (Q712, Vazyme, China) and QuantStudio3 Real-Time PCR System (Quantstudio3, Applied Biosystems, USA). The cyclic parameters used were predenaturation at 95°C for 30 s and PCR (40 cycles of 95°C for 5 s and 60°C for 30 s). The $2^{-\Delta\Delta Ct}$ method was used to analyze the relative mRNA expression, with β-actin serving as the housekeeping gene control. The primers used were purchased from SIYOMICRO. The sequences are shown in Table 1.

## Immunofluorescence staining

Immunofluorescence staining was performed to determine the effect of ectoine on type II collagen synthesis in chondrocytes. The cells were collected and grouped like the RT–qPCR test

**Table 1. Primer sequences for RT–qPCR.**

| Gene | Primer Sequence |
|---|---|
| COX-2 | F:5'–AAAGGCCTCCATTGACCAGA–3' |
| | R:5'–TCGATGTCATGGTAGAGGGC–3' |
| MMP-3 | F:5'–TTTGGCCGTCTCTTCCATCC–3' |
| | R:5'–GCATCGATCTTCTGGACGGT–3' |
| MMP-9 | F:5'–CTACACGGAGCATGGCAACGG–3' |
| | R:5'-TGGTGCAGGCAGAGTAGGAGTG–3' |
| Col2A1 | F:5'– GGCAATAGCAGGTTCACGTACA –3' |
| | R:5'– GATAACAGTCTTGCCCCACTTACC –3' |
| β-actin | F:5'–GGGAAATCGTGCGTGAC–3' |
| | R:5'–TTGCCAATGGTGATGACCTG–3' |

COX: cyclooxygenase-2; MMP-3, -9: matrix metalloproteinase-3, -9

Col2A1: collagen type II alpha 1

to establish a chondrocyte OA model. The chondrocytes in each group were fixed with 4% formaldehyde for 30 min, permeabilized with 0.2% Triton X-100 (T8787, Sigma, USA) for 5 min, and blocked with 5% BSA for 20 min. Then, a primary antibody against type II collagen (1:120, 28459-1-AP, Proteintech, USA) was added at 4°C, after which the mixture was incubated overnight. Thereafter, the FITC-labeled secondary antibody (1:500; SA00003-2; Proteintech, USA) was added to the cells, which were incubated at room temperature for 30 min, blocked with 95% glycerin, and finally observed by a fluorescence microscope (CKX-FL-1; Olympus, Japan). The fluorescence intensity was measured by ImageJ software.

## Histological assessment

The rats were sacrificed at 8 w postsurgery, and tibial specimens were collected from the knee joints. For histological assessment, knee joint specimens were fixed in 4% paraformaldehyde for 24 h and then decalcified in 10% ethylenediaminetetraacetic acid (EDTA) solution for 3–4 w. The samples were subsequently dehydrated, embedded in paraffin, and sectioned into 5-μm coronal sections for further experiments. The selected sections were stained with hematoxylin and eosin (H-E) and Safranin O-fast green (S-O). Thereafter, to determine the severity of joint cartilage damage, cartilage and subchondral bone cellularity and morphology were evaluated using the Osteoarthritis Research Society International (OARSI) scoring system following a previously reported protocol [15].

## Statistical analysis

All experiments were repeated at least three times, and the experimental results were statistically analyzed using GraphPad Prism 9 (GraphPad Software, USA) software. All the data are expressed as the mean ± standard deviation. One-way ANOVA was performed to analyze the RT–qPCR data, cell viability data, immunofluorescence data, and OARSI scores. The *tukey* test was used for pairwise comparisons between multiple groups. $P < 0.05$ was considered to indicate statistical significance.

## Results

### Ectoine improved the trypsin resistance of chondrocytes

Fig 2 shows the effect of different concentrations of ectoine on the trypsin resistance of chondrocytes. In the initial state, the chondrocytes were connected to form a single layer. After treatment with 0.25% trypsin for 2 min, the chondrocytes in the blank control group had contracted and turned into spot-like cells. There were fewer spot-like cells in the 0.5% ectoine group and significantly fewer in the 1.0% ectoine group, and most of the cells were spindle-shaped. The chondrocyte morphology in the 1.5% ectoine group did not change significantly with trypsin treatment [S1 File]. These results suggested that ectoine exerted a protective effect on the trypsin resistance of chondrocytes, and this effect was positively correlated with the ectoine concentration.

### Ectoine improved cell viability at 50°C

The effect of ectoine on the heat resistance of chondrocytes was assessed by the MTT assay. The cell viability of chondrocytes treated with different concentrations of ectoine at 50°C was 68%, 83%, and 89%, respectively, and these values significantly differed from those of the experimental control group (38%, $P < 0.001$). Cell viability was greater in the 1.0% ectoine group than in the 0.5% ectoine group ($P = 0.0031$), but no significant difference was observed between the 1.0% ectoine and 1.5% ectoine groups ($P = 0.41$). Nevertheless, as the ectoine

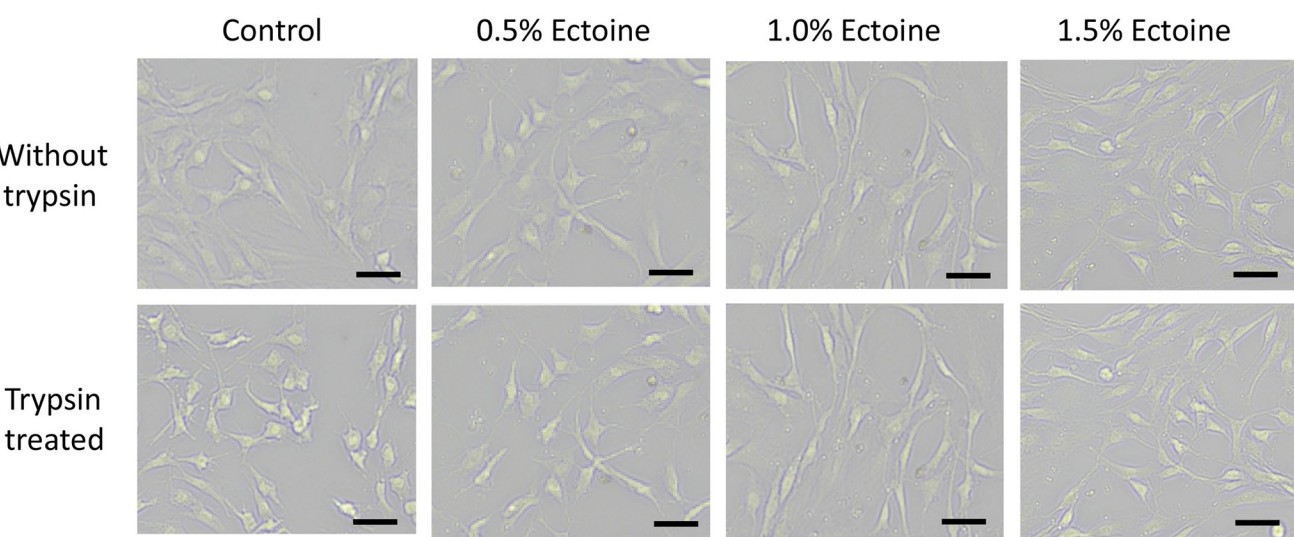

**Fig 2. Effect of ectoine on trypsin resistance in chondrocytes.** With increasing Ectoine concentration, fewer chondrocytes contracted into spot-like cells after treatment with trypsin. (n≥3, scale bar: 10 μm).

concentration increased, the high-temperature tolerance of the chondrocytes gradually increased (Fig 3) [S1 Table].

## Ectoine reduced ROS in chondrocytes

The antioxidant effects of ectoine and dexamethasone (DEX) on chondrocytes were investigated using a ROS assay (Fig 4A). The mean fluorescence intensity in the chondrocytes of the DEX group and different concentrations of ectoine group was significantly lower than that in the experimental control group ($H_2O_2$ group) but greater than that in the blank control group. The mean fluorescence intensities of the 0.5% and 1.0% ectoine groups were not significantly different from that of the DEX group (P = 0.58), while the fluorescence intensity of the 1.5% ectoine group was significantly lower than that of the DEX group (P = 0.003) (Fig 4B), indicating that ectoine has a similar or even better antioxidative effect than DEX and reduces the ROS level in cells [S2 File].

## Ectoine inhibited IL-1β-induced COX-2, MMP-3, and MMP-9 expression in chondrocytes

Chondrocytes were pretreated with different concentrations of ectoine or DEX followed by IL-1β treatment, and COX-2, MMP-3, and MMP-9 gene expression was significantly upregulated in the experimental control group (IL-1β group) compared to the blank control group (P < 0.001). This suggested the successful establishment of the OA model. COX-2 (Fig 5A), MMP-3 (Fig 5B), and MMP-9 (Fig 5C) were significantly downregulated in each of the ectoine groups (P < 0.001) and were positively correlated with the ectoine concentration [S3 Table]. Among these three genes, no significant difference was detected between the DEX and 1.5% ectoine groups; the difference in expression between the 1.5% ectoine group and the blank control group was the smallest, indicating that 1.5% ectoine had the most notable inhibitory effect on inflammation.

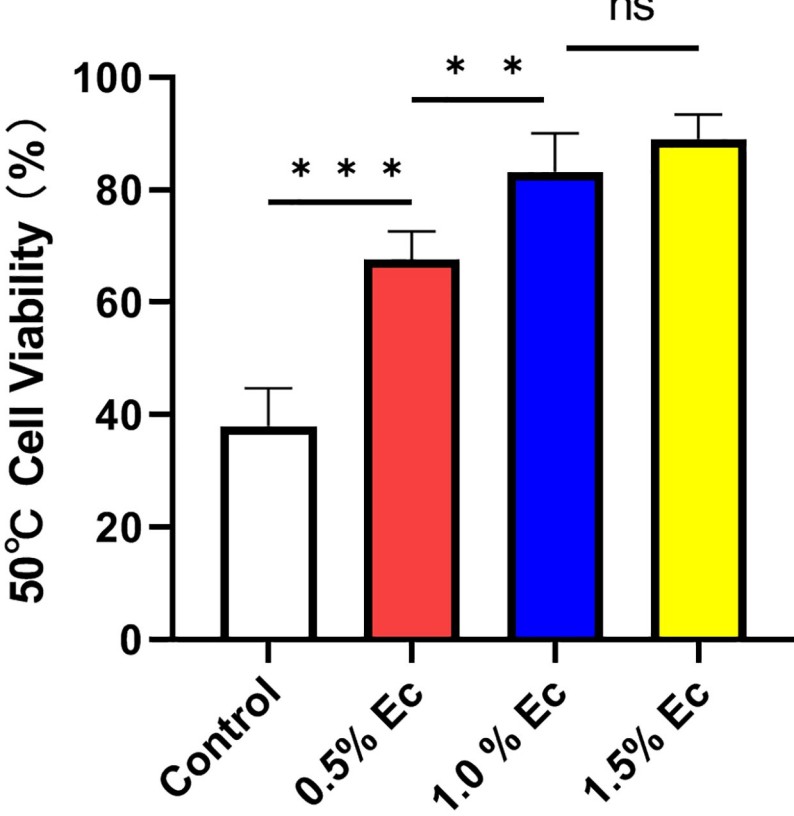

**Fig 3. Cell viability of chondrocytes at 50°C determined by MTT assay.** As the ectoine concentration increased, the high-temperature tolerance of the chondrocytes gradually increased. (n≥6, one-way ANOVA, **P < 0.01, ***P < 0.001).

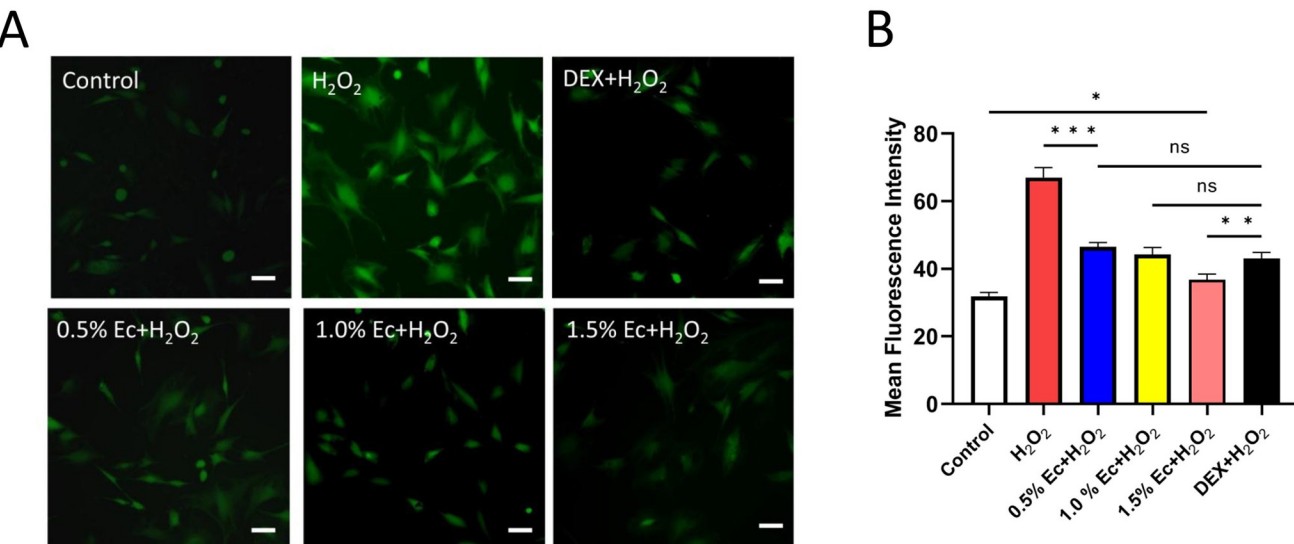

**Fig 4.** ROS assay (A) Images of ROS in chondrocytes pretreated with ectoine or dexamethasone (DEX) and poststimulated with $H_2O_2$. (B) The mean fluorescence intensity in chondrocytes from the different concentrations of ectoine group was significantly lower than that in chondrocytes from the $H_2O_2$ group, and the fluorescence intensity in chondrocytes from the 1.5% ectoine group was significantly lower than that in chondrocytes from the DEX group. (n≥3, one-way ANOVA, scale bar: 10 μm, *P < 0.05, **P < 0.01, ***P < 0.001).

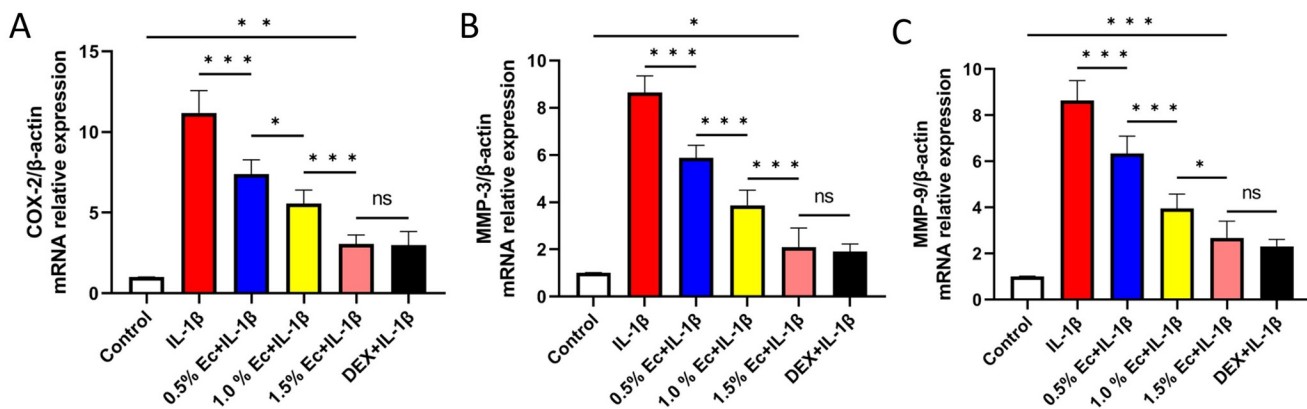

**Fig 5. COX-2, MMP-3, and MMP-9 relative expression in chondrocytes.** (A-C) Chondrocytes were pretreated with different concentrations of ectoine or DEX and then treated with IL-1β. COX-2, MMP-3, and MMP-9 were significantly downregulated in each ectoine group and were positively correlated with the ectoine concentration. (n≥3, one-way ANOVA, *P < 0.05, ***P < 0.001).

### Ectoine maintained type II collagen synthesis in chondrocytes

The effect of ectoine on type II collagen synthesis in chondrocytes was investigated by immunofluorescence staining (Fig 6A). The fluorescence intensities of type II collagen in chondrocytes in the DEX and ectoine groups treated with different concentrations of ectoine were greater than those in the experimental control group (IL-1β group) but lower than those in the blank control group (P < 0.01). Moreover, the fluorescence intensity of each ectoine group was positively correlated with the solution concentration; however, the fluorescence intensity of only the 1.5% ectoine group was significantly different from that of the DEX group (P < 0.001) (Fig 6B). RT–qPCR was used to detect Col2A1 gene expression in each group (Fig 6C). The results suggested that the expression of the Col2A1 gene in the chondrocytes from the DEX and different concentrations of ectoine groups was greater than that in the experimental control group but weaker than that in the blank control group (P < 0.001). Similarly, only the fluorescence intensity of the 1.5% ectoine group was significantly different from that of the DEX group (P = 0.05), indicating that ectoine exerts a similar or even stronger anti-inflammatory effect than DEX and reduces inflammatory reactions in cells [S3 File].

### Ectoine alleviated the development of OA in the rat model

The protective effect of ectoine in a surgically induced rat model was also studied *in vivo*. H-E and S-O staining were performed for histological analysis of the cartilage. Compared with those in the sham group, the OA group exhibited apparent cartilage abrasion, noticeable cartilage hypocellularity, and extensive proteoglycan loss. In contrast, smoother cartilage surfaces and lessened proteoglycan loss were observed after ectoine or HA treatment (Fig 7A) [S4 File]. The OARSI scores in the OA group were greater than those in the sham group, while the OARSI scores were lower in the ectoine group than in the OA group. In addition, no significant difference in OARSI score was observed between the OA group and HA group (P = 0.92) (Fig 7B). These findings suggested that ectoine relieves surgery-induced OA by slowing articular cartilage degeneration.

## Discussion

Herein, the effects of ectoine on trypsin resistance, heat resistance, antioxidation in chondrocytes, and anti-inflammatory effects on IL-1β-induced and surgery-induced OA were

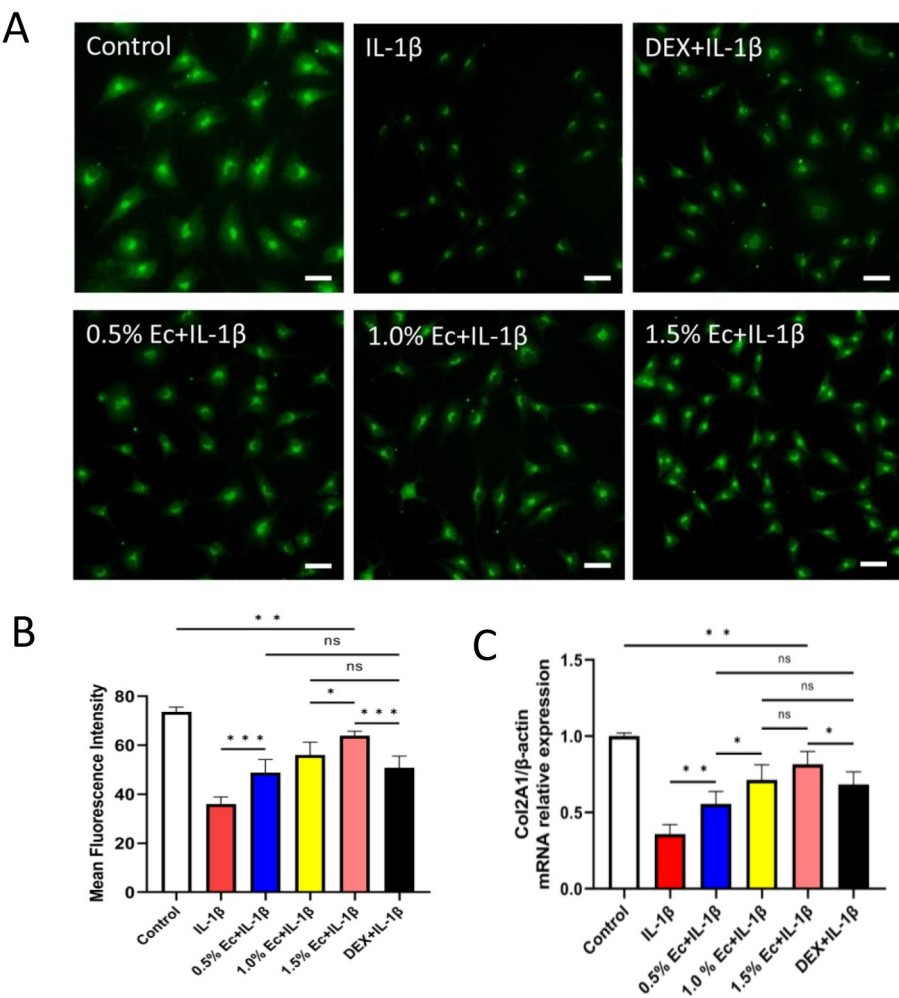

**Fig 6. Immunofluorescence staining and RT–qPCR analysis of type II collagen.** (A). Immunofluorescence staining images of chondrocytes pretreated with ectoine or dexamethasone (DEX) and post-stimulated with IL-1β. (B) The fluorescence intensities of type II collagen in chondrocytes in the ectoine groups treated with different concentrations of ectoine were greater than those in the IL-1β group, but only the fluorescence intensity in the 1.5% ectoine group was significantly different from that in the DEX group. (C) The expression of Col2A1 in the different groups. (n≥3, one-way ANOVA, scale bar: 10 μm, *P < 0.05, **P < 0.01, ***P < 0.001).

investigated. The results suggested that ectoine significantly increased the trypsin resistance of chondrocytes. No noticeable morphological changes were observed in the chondrocytes after 2 min of trypsin treatment. In addition, ectoine retained the viability of chondrocytes at 50°C, improved chondrocyte resistance to oxidation, and reduced intracellular ROS. Compared with IL-1β treatment alone, ectoine pretreatment significantly reduced COX-2, MMP-3, and MMP-9 expression in chondrocytes, maintaining the synthesis of type II collagen and Col2A1 expression and demonstrating a decent anti-inflammatory effect. Furthermore, intra-articular injection of ectoine effectively slowed the degeneration of articular cartilage, thus demonstrating the potential of this approach for OA treatment.

Currently, trypsin is the most widely used digestive reagent in cell culture. It acts on peptide tendons connected with lysine or arginine to remove intercellular mucin and glycoprotein, affects the cytoskeleton, and degrades proteins on the cell membrane at the junction with the

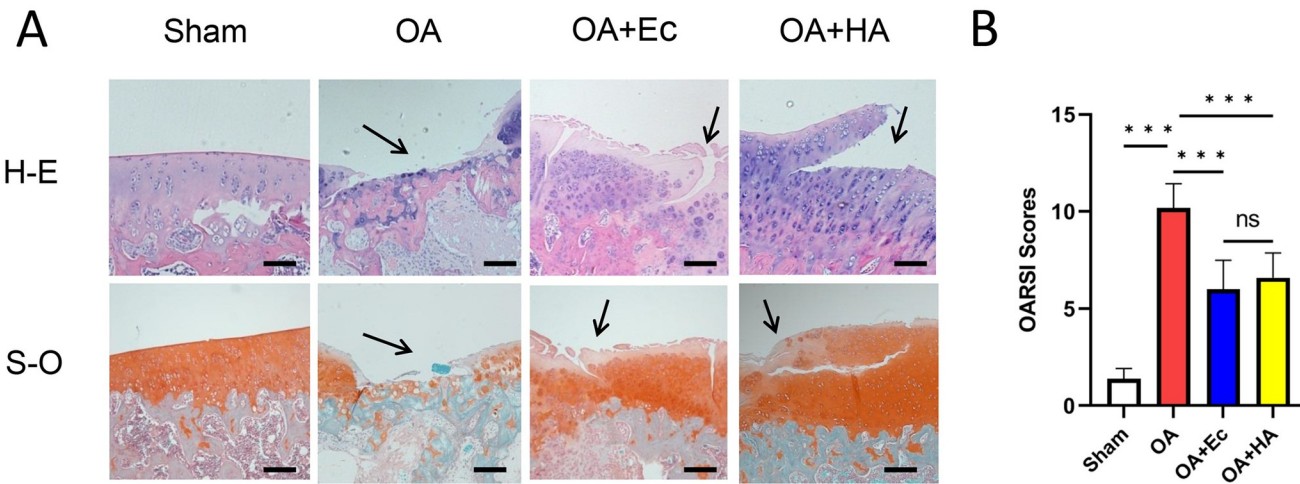

**Fig 7. Effect of ectoine on OA development in the rat model.** (A) Typical H-E and S-O staining of cartilage and subchondral cortical bone from the different experimental groups at 8 w postsurgery. (B) Diagram showing the OARSI scores of the cartilage. (n = 10, one-way AVOVA, scale bar: 100 μm, ***P < 0.001).

wall of the culture dish, thereby separating the cells [16]. Ectoine can improve cell resistance to extreme environments by increasing the fluidity of the lipid head groups of the cell membrane and improving the hydration of the cell surface.

In the present study, chondrocytes contracted into spot-like cells when subjected to trypsin treatment, while the conversion of chondrocytes pretreated with different concentrations of ectoine into spot-like cells following trypsin treatment was significantly reduced, and a positive correlation was observed with the ectoine concentration. These results suggested that ectoine protects chondrocytes from trypsin stimulation and slows digestion. In addition, ectoine protects macromolecules from proteolytic agents. For example, ectoine reduces the activity of the proteolytic enzyme trypsin so that chymotrypsin can resist the influence of trypsin and reduce its activation [17]. The observed dose-dependent effect of ectoine on trypsin resistance in chondrocytes may be attributed to a combination of receptor-mediated signaling, physical barrier formation, modulation of gene expression, and nonspecific interactions with trypsin.

Given that the local temperature of the cartilage surface increases owing to repeated friction and increased pressure during joint movements, particularly after strenuous exercise, a high-temperature resistance experiment was designed for chondrocytes. High temperature notably influences cell activity, primarily by increasing cell membrane permeability and affecting cell membrane fluidity. For example, heatstroke can cause nerve cell and muscle tissue necrosis and damage liver and kidney function. Gao et al. [18] reported that heating primary cultured rat striatal neurons at 43°C for 1 h resulted in abnormal cell membrane phospholipid metabolism and decreased cell membrane fluidity, which affected cell membrane function.

Ectoine is isolated from heat-resistant halophilic bacteria. Therefore, the protective effect of ectoine on chondrocytes was observed at high temperatures. In this study, 68%, 83%, and 89% of the chondrocytes treated with different concentrations of ectoine at 50°C were viable, which was significantly greater than that of the experimental control group (38%). With increasing ectoine concentration, the resistance of chondrocytes to high temperature gradually increased, and cell viability improved. One possible explanation for this observation is that ectoine may interact with cellular components to stabilize them and prevent heat-induced damage. Ectoine may also regulate the expression of heat shock proteins or other stress-related genes to increase

cellular tolerance to heat stress. Additionally, ectoine may form a protective layer around chondrocytes, acting as a physical barrier that shields cells from heat-induced damage. This protective effect suggested that ectoine may be able to protect chondrocytes during strenuous exercise or in high-temperature environments. Notably, Parwata et al. [19] reported that Ectoine maintains 80% lipase activity at 95˚C. In addition, Ectoine is widely used in anti-aging cosmetics and moisturizing creams owing to its high-temperature protective effect, which can effectively moisturize the skin for a long time and prevent skin dehydration [20].

ROS are a series of reactive oxygen species produced by aerobic cells during metabolic processes and include $^1O_2$, $O_3$, OH•, $H_2O_2$, and $O_2$•- [21]. Increased ROS levels are partly responsible for several diseases, such as tumors, heart disease, osteoarthritis, rheumatoid arthritis, diabetes, Alzheimer's disease, and Parkinson's disease [22]. Previous studies have shown that considerable ROS production leads to oxidative stress, resulting in oxidative damage when excess ROS cannot be properly removed. $H_2O_2$, which is the primary ROS, directly or indirectly causes cellular damage and induces cell death [23]. Therefore, $H_2O_2$ causes oxidative damage by stimulating cells to produce excessive ROS. DEX exerts a good antioxidative effect [24], and we hypothesized that ectoine could reduce the ROS generated by $H_2O_2$ stimulation, thereby exerting a protective effect on cells.

Therefore, ROS assays were performed to evaluate the effects of different concentrations of ectoine and DEX on the antioxidative effects of chondrocytes. The results suggested that the fluorescence intensity of chondrocytes in the different concentrations of ectoine groups was lower than that in the experimental control group ($H_2O_2$ group), and the fluorescence intensity in the 1.5% ectoine group was even lower than that in the DEX group. These results indicated that ectoine exerts similar or even better antioxidative effects than DEX, which reduces ROS levels in cells and maintains cellular activities.

Chondrocytes are the only cells in cartilage tissue, and three factors, abnormal metabolism, inflammation, and cell-matrix degradation, are crucial for the development of OA [25]. IL-1β, an inflammatory cytokine, induces chondrocyte apoptosis, inhibits cartilage anabolism, enhances catabolism, and destroys the metabolic balance of chondrocytes in OA. Therefore, it is an important factor in the occurrence of OA and is often used as an inflammatory factor to induce OA in *in vitro* models [26, 27].

COX-2 is an inducible cyclooxygenase that is generally either not expressed in normal tissues or expressed at very low levels, but its expression can be induced by inflammation and several pathological processes. Previous studies have suggested that the COX-2 gene is crucial for cartilage lesions in OA [28, 29] and that the COX-2 gene is expressed in synovial cells in OA and involved in the entire pathological process [30].

Matrix metalloproteinases (MMPs) degrade the extracellular matrix and are crucial for OA progression. Among them, MMP-3 and MMP-9 are closely related to OA. The MMP-9 expression level is directly proportional to the degree of OA, and MMP-9 can hydrolyze extracellular matrix components such as collagen, fibrin, and elastin, destroy subchondral bone, and aggravate joint injury. Like collagen peptidase, MMP-3 facilitates the degradation of proteoglycans, elastin, type IV collagen, and the basement membrane, causing articular cartilage damage. In addition, MMP-3 accelerates cartilage destruction by activating other MMP members via a cascade-amplifying effect. The longer the course of OA, the more MMPs were present in the synovial membrane [31].

Ectoine exerts an inhibitory effect on inflammatory reactions. It effectively alleviates inflammation in various diseases, such as experimentally induced colitis in rats or nanoparticle-induced neutrophilic lung inflammation, and is a new strategy for treating inflammatory bowel disease and lung inflammation [11]. In addition, ectoine protects the ileal mucosa from ischemia-reperfusion injury, a common complication after small bowel transplantation [32].

Several recent studies reported that ectoine-containing sprays effectively treat acute pharyngitis, allergic rhinitis, and rhinoconjunctivitis [33, 34]. However, the effects of ectoine on OA treatment have not been reported to date.

Herein, the COX-2, MMP-3, and MMP-9 expression levels in IL-1β-treated chondrocytes were significantly greater than those in the blank control group, indicating that the chondrocyte OA model was successfully established. Expressions of COX-2, MMP-3, and MMP-9 expression were significantly downregulated in chondrocytes pretreated with ectoine, suggesting that ectoine reduces the inflammatory reactions of chondrocytes. Immunofluorescence staining of type II collagen was performed to verify the synthesis functions of the chondrocytes. Chondrocytes produce type II collagen, which is the most important component in maintaining the structure and tensile strength of cartilage tissues [35]. Immunofluorescence staining revealed that the fluorescence intensity of type II collagen and that of Col2A1 in chondrocytes from the ectoine groups were greater than those in chondrocytes from the experimental control group (IL-1β group) and were positively correlated with the solution concentration. These findings suggested that the amount of type II collagen in the ectoine-pretreated groups was greater than that in the IL-1β group. DEX is a commonly used anti-inflammatory drug that reduces immune reactions and capillary permeability [36]. The fluorescence intensity of type II collagen in the cells pretreated with 1.5% ectoine was greater than that in the DEX group, indicating that ectoine exerted a similar or even stronger anti-inflammatory effect than DEX and could reduce inflammatory reactions in cells. The possible mechanisms are as follows: ectoine may directly interact with the transcription factors that regulate the expression of COX-2, MMP-3, and MMP-9, and ectoine may interfere with the signaling pathways that lead to the activation of these enzymes. This is why adding ectoine significantly reduced the expression of inflammatory cytokines and the degradation of type II collagen.

In the *in vivo* study, the protective effect of ectoine on cartilage was evaluated using the OARIS scoring system. Intra-articular HA is a safe, effective, and minimally invasive therapeutic tool for treating OA [37]; therefore, we used HA as a positive control. The results indicated that the cartilage structure, chondrocyte number, S-O staining, and tide line integrity were lower in the ectoine and HA groups than in the OA group. Safranin O specifically binds proteoglycans in the cartilage matrix. The results of this study showed that, compared with that in the OA group, the loss of S-O staining in the cartilage matrix in the ectoine and HA groups was significantly lower, indicating that ectoine can effectively inhibit the loss of the cartilage matrix during OA development, as can HA.

Despite providing some valuable insights, this study has several limitations. In *in vitro* studies, Ectoine was confirmed to downregulate COX-2, MMP-3, and MMP-9 expression and maintain Col2A1 expression at the gene level; however, owing to a lack of funding, these effects were not verified at the protein level using western blotting. Moreover, in the *in vivo* studies, the proteins that reflect cartilage injury were not analyzed by immunohistochemistry. Only the preliminary results indicate that ectoine has potential application in cartilage repair and OA treatment, but the specific mechanism of action still needs to be studied.

In conclusion, the findings of this study revealed that ectoine protected chondrocytes by maintaining cell morphology, increasing trypsin resistance, increasing high-temperature resistance, and resisting oxidative stimulation. In addition, it reduced COX-2, MMP-3, and MMP-9 expression to alleviate the inflammatory reaction induced by IL-1β and relieve surgery-induced OA by slowing articular cartilage degeneration. The findings of this study provide a potential target and theoretical basis for OA prevention and treatment and reveal that ectoine is valuable for cartilage injury repair, tissue engineering, and OA treatment. For example, ectoine can be combined with other growth factors or drugs to form a composite material that can be used as a slow-release carrier for cartilage injury repair. It can also be used as a scaffold

material for tissue engineering to support cell growth and differentiation. However, the clinical effectiveness of ectoine and its underlying mechanisms need to be further studied.

## Supporting information

**S1 File. Morphological changes of cells.**
(PDF)

**S2 File. ROS images.**
(PDF)

**S3 File. Collagen-II IF images.**
(PDF)

**S4 File. H-E and S-O staining of cartilage.**
(PDF)

**S1 Table. Cell viability.**
(XLSX)

**S2 Table. ROS IF intensity.**
(XLSX)

**S3 Table. RT-qPCR data.**
(XLSX)

**S4 Table. Collagen-II IF intensity.**
(XLSX)

**S5 Table. Collagen-II RT-qPCR data.**
(XLSX)

**S6 Table. OARIS scores.**
(XLSX)

## Acknowledgments

The authors thank to Dr. Shan Zhan, Dr. Yongli Chen and Lihong Gu for providing ectoine and technology supports, thank Longgang District People's Hospital of Shenzhen for facilitating this research.

## Author Contributions

**Conceptualization:** Peng Li, Yong Huang, Zhiqi Zhu.

**Data curation:** Peng Li, Yong Huang, Lishuai Miao, Zhiqi Zhu.

**Formal analysis:** Peng Li, Yong Huang.

**Funding acquisition:** Peng Li, Yong Huang.

**Investigation:** Peng Li.

**Methodology:** Peng Li, Yong Huang, Lishuai Miao, Zhiqi Zhu.

**Project administration:** Peng Li, Yong Huang.

**Resources:** Peng Li, Lishuai Miao.

**Software:** Peng Li, Yong Huang.

**Supervision:** Zhanjun Shi.

**Validation:** Peng Li.

**Visualization:** Peng Li.

**Writing – original draft:** Peng Li, Yong Huang, Lishuai Miao, Zhiqi Zhu.

**Writing – review & editing:** Peng Li, Yong Huang, Zhanjun Shi.

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
