## [Decision Letter · Decision Letter 0]

27 Nov 2023

PONE-D-23-31640Protective effects of ectoine on articular chondrocytes and cartilage in ratsPLOS ONE

Dear Dr. Li,

Thank you for submitting your manuscript to PLOS ONE. After careful consideration, we feel that it has merit but does not fully meet PLOS ONE’s publication criteria as it currently stands. Therefore, we invite you to submit a revised version of the manuscript that addresses the points raised during the review process.

We look forward to receiving your revised manuscript.

Kind regards,

Bijay Kumar Behera, Ph.D.

Academic Editor

PLOS ONE

Journal Requirements:

"The work was supported by Shenzhen Longgang District Medical Science and Technology Project [grant no. LGWJ 2021-037]."

Additional Editor Comments:

According to the reviewers recommendations, my decision is to major revision of the manuscript.

Reviewers' comments:

Reviewer's Responses to Questions

**Comments to the Author**

1. Is the manuscript technically sound, and do the data support the conclusions?

Reviewer #1: Yes

Reviewer #2: Partly

2. Has the statistical analysis been performed appropriately and rigorously? 

Reviewer #1: Yes

Reviewer #2: Yes

3. Have the authors made all data underlying the findings in their manuscript fully available?

Reviewer #1: Yes

Reviewer #2: No

4. Is the manuscript presented in an intelligible fashion and written in standard English?

Reviewer #1: Yes

Reviewer #2: Yes

5. Review Comments to the Author

Reviewer #1: Addressing osteoarthritis (OA), characterized by articular cartilage degradation and inflammation. In vitro experiments assessed the impact of ectoine on chondrocytes' trypsin resistance, viability at elevated temperatures, and resistance to oxidative stress. The study also examined ectoine's effects on chondrocytes treated with IL-1β, measuring gene expression levels of key markers and assessing type II collagen synthesis. In an in vivo rat OA model, ectoine demonstrated protective effects, reducing cartilage degeneration and Osteoarthritis Research Society International (OARSI) scores. Overall, the findings suggest that ectoine may serve as a potential therapeutic agent for OA by safeguarding chondrocytes and preserving cartilage integrity. I recommend the manuscript for the publication but it require to answer certain questions.

Why does articular cartilage have a weak self-repairing ability after injury, and what are the limitations of current treatments such as debridement, lavage, microfracture, and autologous chondrocyte implantation?

In what ways does the study suggest that ectoine could be beneficial in the field of cartilage repair and OA treatment? Include this in the introduction part.

Experimental Design and Methods:

a. How were the chondrocytes initially prepared and treated with trypsin in the in vitro study?

b. Can you provide more details on the MTT assay used to assess the effect of ectoine on the heat resistance of chondrocytes?

c. What specific ROS species were measured in the ROS assay, and how were they quantified?

d. Were there any control groups in the in vitro experiments to validate the specificity of the observed effects of ectoine?

e. Could you elaborate on the rationale for choosing 50 °C as the temperature for the high-temperature resistance experiment?

Results and Data Interpretation:

a. In Figure 2, what might explain the difference in chondrocyte morphology between the ectoine groups after trypsin treatment?

b. Can you discuss the potential reasons for the observed dose-dependent effect of ectoine on trypsin resistance in chondrocytes?

c. How do you interpret the variations in cell viability at 50 °C with different concentrations of ectoine in Figure 3?

d. Regarding the ROS assay (Figure 4), what implications do the differences in fluorescence intensity between ectoine groups and the DEX group have for the study's objectives?

e. What are the possible mechanisms behind the observed inhibitory effects of ectoine on COX-2, MMP-3, and MMP-9 expression in chondrocytes (Figure 5)?

Future Directions and Implications:

a. In the discussion, you mention potential applications of ectoine in cartilage injury repair and tissue engineering. Can you elaborate on specific strategies or technologies where ectoine could be integrated?

b. What further studies or experiments would you recommend to elucidate the specific mechanisms through which ectoine exerts its protective effects on chondrocytes?

c. How might the findings of this study inform the development of novel therapeutic interventions for OA, and what are the potential challenges in translating these findings to clinical practice?

Reviewer #2: 1. Please correct the language of the paper. There are a few grammatical errors.

2. Figure 7 numbering is missing.

3. In Figure 1 legend, part B, please make changes in the title (Mechanism of ectoine). It should be clear from the title what mechanism is the figure demonstrating.

4. Please make it clear in the figure 2 about the images taken in the presence and absence of trypsin for comparison.

5. In figure 3, it would be better to also add a control group under normal physiological conditions showcasing the normal cell viability. Why only one temperature is tested? Explain the significance.

6. In figure 4 what all percentages of Ectoine were tested? What is the optimum percentage?

7. In all the figures, there is no mention of different parts of the figure (ex: Figure 5A, B, C). Please rectify that in the whole manuscript and in the legend.

8. In figure 7, there are no markings in the image showcasing the difference between the groups.

9. Please explain why other in-vivo studies were not done. Why ROS levels and other in-vitro studies, not shown in rat model as well?

10. In figure 7 (Rat model study), please show the control and positive control as well.

11. Please modify the title as well. It should be clear from the title that the undertaken study is in reference to osteoarthritis.

6. PLOS authors have the option to publish the peer review history of their article (what does this mean?). If published, this will include your full peer review and any attached files.

Reviewer #1: No

Reviewer #2: **Yes: **ANOUSHKA KHANNA

---

## [Author Response · Author response to Decision Letter 0]

17 Dec 2023

Reviewer 1#

Addressing osteoarthritis (OA), characterized by articular cartilage degradation and inflammation. In vitro experiments assessed the impact of ectoine on chondrocytes' trypsin resistance, viability at elevated temperatures, and resistance to oxidative stress. The study also examined ectoine's effects on chondrocytes treated with IL-1β, measuring gene expression levels of key markers and assessing type II collagen synthesis. In an in vivo rat OA model, ectoine demonstrated protective effects, reducing cartilage degeneration and Osteoarthritis Research Society International (OARSI) scores. Overall, the findings suggest that ectoine may serve as a potential therapeutic agent for OA by safeguarding chondrocytes and preserving cartilage integrity. I recommend the manuscript for the publication but it require to answer certain questions.

Why does articular cartilage have a weak self-repairing ability after injury, and what are the limitations of current treatments such as debridement, lavage, microfracture, and autologous chondrocyte implantation?

Answer: Articular cartilage has a weak self-repairing ability after injury due to its limited ability to regenerate. Cartilage tissue lacks a direct blood supply and is not well-supplied with nerves, which makes it difficult for the tissue to repair itself after injury. Additionally, chondrocytes, the primary cells in cartilage tissue, have a low metabolic activity, further limiting the ability of the tissue to repair damage.

Current treatments for articular cartilage injuries include debridement, lavage, microfracture, and autologous chondrocyte implantation. Debridement involves removing the damaged tissue to reduce pain and inflammation, but it does not promote cartilage regeneration. Lavage involves washing out the injured area to remove debris and irritants, but it also does not promote cartilage repair. Microfracture is a surgical technique that creates small fractures in the bone underlying the cartilage injury to stimulate the growth of cartilage-like tissue. However, the resulting tissue is often fibrocartilage, which is not as strong or durable as normal articular cartilage. Autologous chondrocyte implantation involves harvesting chondrocytes from a patient's own healthy cartilage and growing them in the laboratory to form new cartilage tissue, which is then implanted into the injured area. While this treatment can produce good results in some patients, it is expensive, time-consuming, and requires cell harvesting from a healthy part of the joint, which can lead to further damage.

In what ways does the study suggest that ectoine could be beneficial in the field of cartilage repair and OA treatment? Include this in the introduction part.

Answer: The study suggested that ectoine could be beneficial in the field of cartilage repair and OA treatment in several ways. First, ectoine has anti-inflammatory properties that can help reduce cartilage damage caused by inflammation. Second, it can help protect chondrocytes from damage caused by oxidants and reactive oxygen species, which are known to play a role in OA development. Finally, ectoine promoted cartilage matrix synthesis and help restore the structure and function of cartilage tissue.

Those were included in the introduction part of revised manuscript.

Experimental Design and Methods:

a. How were the chondrocytes initially prepared and treated with trypsin in the in vitro study?

Answer: In the in vitro study, chondrocytes were initially prepared by harvesting cartilage tissue from rats undergoing knee surgery. The tissue was then minced and digested with trypsin and type II collagenase, to release the chondrocytes from the extracellular matrix. The chondrocytes were then washed and cultured in a medium. When the second-passage chondrocytes had grown to a density of 50%, 1 mL of 0.5%, 1.0%, or 1.5% (w/v) ectoine was added to the wells. Cells not treated with ectoine were used as a blank control group. Three duplicate wells were used for each group. When the cell density reached 80%, 50 µL of 0.25% trypsin was added to digest the cells for 2 min. Finally, the cell morphology was observed.

b. Can you provide more details on the MTT assay used to assess the effect of ectoine on the heat resistance of chondrocytes?

Answer: In this study, the MTT assay was used to assess the effect of ectoine on the heat resistance of chondrocytes. When the cells reached a density of 50%, 100 µL of 0.5%, 1.0%, or 1.5% ectoine solution was added to the wells. Cells not subjected to ectoine treatment were used as the experimental control group. Simultaneously, zero-well and control well were used. After the cells reached 80% confluence, the plants in the ectoine and experimental control groups were incubated at 37°C and 50°C for 1 h, respectively, while the control wells were incubated at 37°C only. The cells were subsequently incubated with 10 µL of MTT at 37°C for 4 h, after which 100 µL of DMSO was added to each well, and the mixture was shaken for 10 min. The OD of each well was measured at 490 nm using a microplate reader.

The MTT reagent is converted by living cells into a purple formazan product, which can be measured spectrophotometrically. The amount of formazan produced is proportional to the number of living cells, and thus, it provides a quantitative measure of cell viability. By comparing the viability of chondrocytes treated with ectoine to those not treated with ectoine after heat stress exposure, the study was able to assess the protective effect of ectoine on chondrocyte heat resistance.

c. What specific ROS species were measured in the ROS assay, and how were they quantified?

Answer: In the ROS assay, the specific ROS species measured were hydrogen peroxide (H2O2). These ROS species were quantified using imageJ software. The image was converted to grayscale to enhance contrast and eliminate background noise. Thresholding was applied to the image. The software calculates the average intensity of all the pixels within the region of interest and provides a numerical value representing the mean fluorescence intensity. This value can then be quantified and used for further analysis.

d. Were there any control groups in the in vitro experiments to validate the specificity of the observed effects of ectoine?

Answer: Yes, there were control groups in the in vitro experiments. Briefly, cells not treated with ectoine were used as the experimental control group. In addition, dexamethasone group was used as a positive control in the ROS assay, RT-qPCR and immunofluorescence staining test.

e. Could you elaborate on the rationale for choosing 50 °C as the temperature for the high-temperature resistance experiment?

Answer: Choosing 50°C as the temperature for the high-temperature resistance experiment of chondrocytes is based on a comprehensive consideration of safety, reliability, and practicality.

On the one hand, ectoine is isolated from thermophilic bacteria. The optimal cultivation temperature for chondrocytes is 37 °C, while the optimal cultivation temperature for thermophilic bacteria is around 50 °C. Therefore, we believe that conducting a cell heat tolerance experiment at 50 °C with the addition of ectoine can demonstrate its protective effect on chondrocytes at high temperature. 

On the other hand, 50°C is a relatively high temperature that mimics the extreme environment to which cartilage in the human body may be exposed. Cartilage in the body may constantly subjected to high temperatures, so testing its tolerance at this temperature provides an assessment of its reliability under in vivo conditions.

Results and Data Interpretation:

a. In Figure 2, what might explain the difference in chondrocyte morphology between the ectoine groups after trypsin treatment?

Answer: The difference in chondrocyte morphology between the ectoine groups after trypsin treatment may be explained by the protective effect of ectoine on the cells. Ectoine is a stress-protecting amino acid derivative that can protect cells from various types of stress, including heat, freezing, dehydration, and oxidants. In this study, the addition of ectoine to the chondrocyte cultures may have provided protection against the stress caused by trypsin treatment. As the concentration increases, the protective effect of ectoine also increases.

b. Can you discuss the potential reasons for the observed dose-dependent effect of ectoine on trypsin resistance in chondrocytes?

Answer: The observed dose-dependent effect of ectoine on trypsin resistance in chondrocytes may be attributed to multiple factors. One possible explanation is that ectoine may bind to specific receptors on the chondrocyte surface, leading to activation of signaling pathways that promote cell survival and resistance to trypsin. Ectoine's protective effect may also be due to its ability to form a protective layer around the chondrocytes, acting as a physical barrier that interferes with trypsin's access to its targets. Furthermore, ectoine may influence the expression of genes involved in cell survival and defense mechanisms. In particular, genes encoding proteins that contribute to cell wall integrity, autophagy, and antioxidant defense systems may be upregulated in response to ectoine treatment. 

c. How do you interpret the variations in cell viability at 50 °C with different concentrations of ectoine in Figure 3?

Answer: The variations in cell viability at 50 °C with different concentrations of ectoine can be interpreted in terms of the protective effect of ectoine on chondrocytes is concentration-dependent, with higher concentrations providing greater protection against heat-induced stress. 

One possible explanation for this observation is that ectoine may interact with cellular components to stabilize them and prevent heat-induced damage. Ectoine may also regulate the expression of heat shock proteins or other stress-related genes to increase cellular tolerance to heat stress. Additionally, ectoine may form a protective layer around chondrocytes, acting as a physical barrier that shields cells from heat-induced damage.

d. Regarding the ROS assay (Figure 4), what implications do the differences in fluorescence intensity between ectoine groups and the DEX group have for the study's objectives?

Answer: The fluorescence intensities of the 0.5% and 1.0% ectoine groups were not significantly different from that of the DEX group, while the fluorescence intensity of the 1.5% ectoine group was significantly lower than that of the DEX group, indicating that ectoine has a similar or even better antioxidative effect than DEX and reduces the ROS level in cells. 1.5% ectoine was more effective in scavenging ROS or inhibiting their production. This could be attributed to the antioxidant properties of ectoine, which may contribute to its protective effect on chondrocytes.

e. What are the possible mechanisms behind the observed inhibitory effects of ectoine on COX-2, MMP-3, and MMP-9 expression in chondrocytes (Figure 5)?

Answer: The possible mechanisms behind the observed inhibitory effects of ectoine on COX-2, MMP-3, and MMP-9 expression in chondrocytes are not fully understood. However, several possible explanations can be proposed.

Firstly, ectoine may directly interact with the transcription factors that regulate the expression of these enzymes, such as NF-κB or AP-1, and inhibit their activation. This interaction could prevent the binding of these transcription factors to the promoters regions of the COX-2, MMP-3, and MMP-9 genes, resulting in their decreased expression.

Secondly, ectoine may interfere with the signaling pathways that lead to the activation of these enzymes. For example, it could inhibit the phosphorylation of kinases that regulate the activity of transcription factors or prevent the activation of protein kinases responsible for the induction of COX-2 and MMPs.

Finally, ectoine may affect the expression of these enzymes by modifying the redox status of chondrocytes. It has been shown that oxidative stress plays a role in the upregulation of COX-2 and MMPs in chondrocytes, and ectoine has been reported to have antioxidant properties. Therefore, ectoine may prevent the accumulation of reactive oxygen species (ROS) and protect chondrocytes from oxidative damage, thereby reducing the expression of these enzymes.

Future Directions and Implications:

a. In the discussion, you mention potential applications of ectoine in cartilage injury repair and tissue engineering. Can you elaborate on specific strategies or technologies where ectoine could be integrated?

Answer: Ectoine can be integrated into cartilage injury repair and tissue engineering through the following specific strategies and technologies:

1. Cartilage injury repair: Ectoine can be used to protect cartilage cells and promote cartilage regeneration. It can be combined with other growth factors or drugs to form a composite material that can be used as a slow-release carrier for cartilage injury repair. This composite material can be directly implanted into the cartilage injury site to release ectoine and other factors slowly, thereby promoting cartilage regeneration and repair.

2. Tissue engineering: Ectoine can be used as a cell culture medium component to maintain the survival and activity of cells. It can also be used as a scaffold material for tissue engineering to support cell growth and differentiation. When used as a scaffold material, ectoine can be combined with other materials to form a composite scaffold that provides a good microenvironment for cell growth and differentiation, and promotes tissue regeneration and repair.

b. What further studies or experiments would you recommend to elucidate the specific mechanisms through which ectoine exerts its protective effects on chondrocytes?

Answer: To further study the mechanisms through which ectoine exerts its protective effects on chondrocytes, the following experiments and studies can be recommended:

1. Cell culture experiments: chondrocytes can be cultured in vitro in the presence of ectoine, and the expression of chondrocyte-specific genes and proteins can be detected by RT-PCR, Western blot, immunohistochemistry, etc. to determine whether ectoine can affect chondrocyte differentiation and proliferation.

2. Animal model experiments: Histological and immunohistochemical examination of cartilage tissue can be used to evaluate the degree of cartilage injury and repair.

3. Protein array and Western blot analysis: We can detect the expression levels of a variety of signaling pathway proteins in chondrocytes after ectoine treatment, and then use bioinformatics analysis to identify the signaling pathways that are activated or inhibited by ectoine, and then analyze whether ectoine can activate or inhibit these signaling pathways by Western blot analysis.

c. How might the findings of this study inform the development of novel therapeutic interventions for OA, and what are the potential challenges in translating these findings to clinical practice?

Answer: The findings of this study on ectoine could potentially inform the development of novel therapeutic interventions for OA by further understanding the protective effects of ectoine on chondrocytes and its possible mechanisms of action. This knowledge could lead to the development of new drugs or treatments that mimic the effect of ectoine or enhance the efficacy of existing drugs in chondrocytes, providing a novel approach for OA therapy.

The potential challenges in translating these findings to clinical practice includes the translation of preclinical research results to humans and the identification of appropriate outcome measures for assessing the effectiveness of OA therapies. In addition, the cost and duration of clinical trials can be significant barriers to the translation of preclinical research findings to clinical practice. 

Reviewer 2#

1. Please correct the language of the paper. There are a few grammatical errors.

Answer: The paper was edited by a native English-speaking editor at Editeg and a language certificate was provided.

2. Figure 7 numbering is missing.

Answer: The number of figure 7 was added in the revised version.

3. In Figure 1 legend, part B, please make changes in the title (Mechanism of ectoine). It should be clear from the title what mechanism is the figure demonstrating.

Answer: The legend of figure 1 was changed and demonstrated the mechanism directly.

4. Please make it clear in the figure 2 about the images taken in the presence and absence of trypsin for comparison.

Answer: The figure 2 was revised and demonstrated the presence and absence of trypsin.

5. In figure 3, it would be better to also add a control group under normal physiological conditions showcasing the normal cell viability. Why only one temperature is tested? Explain the significance.

Answer: The control group cultured under 37 °C was set in control well and took as a control group in the MTT assay, in that case, the cell viability was 100% under normal physiological condition. That is why we did not show it in the chart. 

Choosing 50°C as the temperature for the high-temperature resistance experiment of chondrocytes was based on a comprehensive consideration of safety, reliability, and practicality. 

On the one hand, ectoine is isolated from thermophilic bacteria. The optimal cultivation temperature for chondrocytes is 37 °C, while the optimal cultivation temperature for thermophilic bacteria is around 50 °C. Therefore, we believe that conducting a cell heat tolerance experiment at 50 °C with the addition of ectoine can demonstrate its protective effect on chondrocytes under high temperature. 

On the other hand, 50°C is a relatively high temperature that mimics the extreme environment to which cartilage in the human body may be exposed. Cartilage in the body may constantly subjected to high temperatures, so testing its tolerance at this temperature provides an assessment of its reliability under in vivo conditions.

This protective effect suggests that ectoine may be able to protect chondrocytes during strenuous exercise or in high-temperature environments.

6. In figure 4 what all percentages of Ectoine were tested? What is the optimum percentage?

Answer: In figure 4, we tested the effects of three percentages of ectoine: 0.5%, 1.0%, and 1.5%. From the results of this experiment, 1.5% was the optimal percentage. Also, during the preliminary experiment, we found that there was no significant difference in the proliferation of chondrocytes between 1.0% and 1.5% ectoine. Of course, perhaps the percentage of 1.5% is not the optimal percentage for ectoine to have antioxidant effects, but the focus of this experiment is to demonstrate the antioxidant effects of ectoine on chondrocytes, and exploring the optimal percentage is the next goal.

7. In all the figures, there is no mention of different parts of the figure (ex: Figure 5A, B, C). Please rectify that in the whole manuscript and in the legend.

Answer: The different parts of the figures were mentioned in the revised manuscript and legend.

8. In figure 7, there are no markings in the image showcasing the difference between the groups.

Answer: Markings were added in the figure 7 to show the difference between the groups.

9. Please explain why other in-vivo studies were not done. Why ROS levels and other in-vitro studies, not shown in rat model as well?

Answer: The main reason for not conducting other in vivo studies is insufficient research funding. Antibodies for in vivo experiments are generally expensive. According to literature review, ROS and RT-qPCR tests are mainly conducted in vitro, while the main validation method for in vivo studies is immunohistochemical tests on some inflammatory cytokines or cartilage-specific proteins. This is also our next research direction.

10. In figure 7 (Rat model study), please show the control and positive control as well.

Answer: in figure 7, the sham group was sued as control group, and the hyaluronic acid group was used as positive control group.

11. Please modify the title as well. It should be clear from the title that the undertaken study is in reference to osteoarthritis.

 Answer: The title was changed to” Protective effects of ectoine on articular chondrocytes and cartilage in rats for treating osteoarthritis.”

---

## [Decision Letter · Decision Letter 1]

9 Feb 2024

Protective effects of ectoine on articular chondrocytes and cartilage in rats for treating osteoarthritis.

PONE-D-23-31640R1

Dear Dr. Li,

We’re pleased to inform you that your manuscript has been judged scientifically suitable for publication and will be formally accepted for publication once it meets all outstanding technical requirements.

Kind regards,

Bijay Kumar Behera, Ph.D.

Academic Editor

PLOS ONE

Additional Editor Comments (optional):

The authors have addressed the reviewers comments. My decision is to accept the manuscript.

Reviewers' comments:

Reviewer's Responses to Questions

**Comments to the Author**

1. If the authors have adequately addressed your comments raised in a previous round of review and you feel that this manuscript is now acceptable for publication, you may indicate that here to bypass the “Comments to the Author” section, enter your conflict of interest statement in the “Confidential to Editor” section, and submit your "Accept" recommendation.

Reviewer #2: All comments have been addressed

2. Is the manuscript technically sound, and do the data support the conclusions?

Reviewer #2: Yes

3. Has the statistical analysis been performed appropriately and rigorously? 

Reviewer #2: Yes

4. Have the authors made all data underlying the findings in their manuscript fully available?

Reviewer #2: Yes

5. Is the manuscript presented in an intelligible fashion and written in standard English?

Reviewer #2: Yes

6. Review Comments to the Author

Reviewer #2: The study seems to provide a new insight for the treatment of OA. Although further experiments are required to reach to a solid conclusion. As of now, it can serve as a starting point and basis for further studies.

7. PLOS authors have the option to publish the peer review history of their article (what does this mean?). If published, this will include your full peer review and any attached files.

Reviewer #2: No

---

## [Editor Report · Acceptance letter]

19 Feb 2024

PONE-D-23-31640R1 

PLOS ONE

Dear Dr. Li, 

I'm pleased to inform you that your manuscript has been deemed suitable for publication in PLOS ONE. Congratulations! Your manuscript is now being handed over to our production team.

Kind regards, 

on behalf of

Dr. Bijay Kumar Behera 

Academic Editor

PLOS ONE